# A multispecies coalescent model for quantitative traits

Fábio K Mendes[1]*, Jesualdo A Fuentes-González[1,2], Joshua G Schraiber[3,4,5], Matthew W Hahn[1,6]

[1]Department of Biology, Indiana University, Bloomington, United States; [2]School of Life Sciences, Arizona State University, Tempe, United States; [3]Department of Biology, Temple University, Philadelphia, United States; [4]Center for Computational Genetics and Genomics, Temple University, Philadelphia, United States; [5]Institute for Genomics and Evolutionary Medicine, Temple University, Philadelphia, United States; [6]Department of Computer Science, Indiana University, Bloomington, United States

**Abstract** We present a multispecies coalescent model for quantitative traits that allows for evolutionary inferences at micro- and macroevolutionary scales. A major advantage of this model is its ability to incorporate genealogical discordance underlying a quantitative trait. We show that discordance causes a decrease in the expected trait covariance between more closely related species relative to more distantly related species. If unaccounted for, this outcome can lead to an overestimation of a trait's evolutionary rate, to a decrease in its phylogenetic signal, and to errors when examining shifts in mean trait values. The number of loci controlling a quantitative trait appears to be irrelevant to all trends reported, and discordance also affected discrete, threshold traits. Our model and analyses point to the conditions under which different methods should fare better or worse, in addition to indicating current and future approaches that can mitigate the effects of discordance.

DOI: https://doi.org/10.7554/eLife.36482.001

*For correspondence:
fkmendes@indiana.edu

Competing interests: The authors declare that no competing interests exist.

## Introduction

Understanding how traits evolve through time is one of the major goals of phylogenetics. Phylogenetic inferences made about traits can include estimating a trait's evolutionary rate and ancestral states, determining whether the evolution of a trait is influenced by natural selection, and establishing whether certain character states make speciation and extinction more or less likely (*Harvey and Pagel, 1991*; *O'Meara, 2012*; *Garamszegi, 2014*). Despite the variety of questions one can ask, and the plethora of different discrete and continuous traits that can be studied, it has long been recognized that in order to make inferences about trait evolution it is imperative to consider how the species carrying these traits are related (*Felsenstein, 1985*). Phylogenetic comparative methods model traits as evolving along a phylogeny, and therefore often require one, or sometimes multiple, species trees as input (*Pagel, 1999*; *O'Meara, 2012*; *Hahn and Nakhleh, 2016*).

The unprecedented increase in the availability of molecular data has been a boon to the construction of well-supported species trees — that is, those with high levels of statistical support. Thanks to advances in sequencing technology, species trees are now denser, taller, and better resolved. In contrast to the high levels of support provided by genome-scale data, phylogenomic studies have also revealed topological discordance between gene trees to be pervasive across the tree of life (*Pollard et al., 2006*; *White et al., 2009*; *Hobolth et al., 2011*; *Brawand et al., 2014*; *Zhang et al., 2014*; *Suh et al., 2015*; *Pease et al., 2016*). Gene trees can disagree with one another and with the species tree because of technical reasons — for example, model misspecification, low phylogenetic

signal, or the mis-identification of paralogs as orthologs — but also as a result of biological phenomena such as incomplete lineage sorting (ILS), introgression, and horizontal gene transfer (*Maddison, 1997*). Among the latter, ILS is well-studied due to its conduciveness to mathematical characterization (*Hudson, 1983*; *Tajima, 1983*; *Pamilo and Nei, 1988*), in addition to being an inevitable result of population processes (*Edwards, 2009*). Going backwards in time, ILS is said to occur when lineages from the same population do not coalesce in that population, but instead coalesce in a more ancestral population. If these lineages then happen to coalesce first with others from more distantly related populations, the gene tree will be discordant with the species tree.

The realization that genealogical discordance is the rule rather than the exception (in species trees with short internal branches) has been followed closely by a growing awareness that failing to consider microevolutionary-scale processes in phylogenetic inferences can be problematic (*Kubatko and Degnan, 2007*; *Edwards, 2009*; *Mendes and Hahn, 2016*; *Mendes and Hahn, 2018*). In the context of species tree estimation, this has led to the development of popular methods that account for processes such as ILS and introgression (e.g. *Liu, 2008*; *Than et al., 2008*; *Liu et al., 2009*; *Heled and Drummond, 2010*; *Larget et al., 2010*; *Bryant et al., 2012*; *Mirarab and Warnow, 2015*; *Solís-Lemus and Ané, 2016*). However, the development of models incorporating discordance in order to deal with trait evolution have lagged behind those estimating species trees.

One way that gene tree discordance can affect inferences about trait evolution is by increasing the risk of hemiplasy. Hemiplasy is the production of a homoplasy-like pattern by a non-homoplastic event (*Avise and Robinson, 2008*), generally because a character-state transition has occurred on a discordant gene tree. Consider the example shown in *Figure 1*: trait one is underlain by a gene whose topology is discordant with the species tree; a single state transition occurs only once along the branch leading to the ancestor of species *A* and *C*. However, if one attempts to infer the history of transitions on the species tree, two spurious transitions (for instance, on branches leading to *A* and *C*) must be invoked. The same occurs with trait two (*Figure 1*), but on the other discordant gene tree. Unless the gene tree underlying a discrete trait is concordant with the species tree (such as trait three in *Figure 1*), ignoring its topology can lead one to believe that homoplasy has happened, when in fact it has not — this is due to hemiplasy.

Recent work on the relevance of gene tree discordance to phylogenetic inferences has demonstrated that hemiplasy is widespread and problematic. At the molecular level, hemiplasy can cause apparent substitution rate variation, can spuriously increase the detection of positive selection in coding sequences, and can lead to artefactual signals of convergence (*Mendes and Hahn, 2016*; *Mendes et al., 2016*). In datasets with high levels of gene tree discordance, the fraction of all substitutions that are hemiplastic can be quite high (*Copetti et al., 2017*). In contrast, the manner in which genealogical discordance might affect studies of complex trait evolution is still not well understood. While past work has investigated how the genetic architecture of complex traits interacts with genetic drift to influence patterns of variation between populations and species (*Lynch, 1988*, *Lynch, 1989*, *Lynch, 1994*; *Whitlock, 1999*; *Ovaskainen et al., 2011*; *Zhang et al., 2014*), an interesting and still unanswered question is whether genealogical discordance can have an effect on these traits.

As continuous traits are often underlain by a large number of loci, a significant fraction of them could have discordant gene trees in the presence of ILS or introgression. Trait-affecting substitutions on discordant internal branches (those that are absent from the species tree; *Mendes and Hahn, 2018*) of such trees would then increase the similarity in traits between more distantly related species, while decreasing that of more closely related species. Such an effect could consequently affect the inferences from phylogenetic comparative methods about these quantitative traits. On the other hand, the most frequent gene tree in a data set is generally expected to be concordant with the species tree (except in cases of anomalous gene trees; *Degnan and Rosenberg, 2006*). As a consequence, we might expect that the contribution to traits from loci with concordant gene trees would outweigh the signal introduced by loci with discordant gene trees, possibly making phylogenetic inferences about continuous traits more robust to hemiplasy relative to discrete traits. In other words, a reasonable hypothesis is that gene tree discordance should only be problematic for traits controlled by a small number of loci, but not for those controlled by many loci (*Hahn and Nakhleh, 2016*).

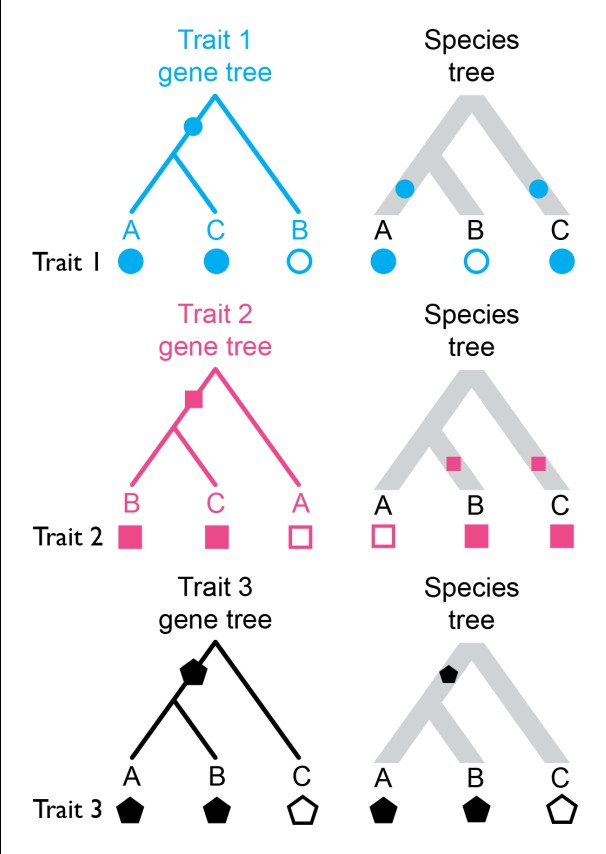

**Figure 1.** Three distinct discrete traits with their states mapped to the gene trees they evolved on, and to the species tree. Hollow and filled shapes represent the ancestral and derived states, respectively, with character-state transitions being indicated by filled shapes along internal branches. Traits 1 and 2 undergo a single character-state transition in their evolutionary history, but when the states are resolved on the species tree, a homoplasy-like (yet not truly homoplastic) pattern emerges (i.e. hemiplasy). Trait three has evolved along a gene tree that matches the species tree in topology, and so no hemiplasy occurs.

DOI: https://doi.org/10.7554/eLife.36482.002

Here, we present a model of quantitative traits evolving under the multispecies coalescent that accounts for gene tree discordance, deriving the expected variances and covariances in quantitative traits under this model. We then apply traditional phylogenetic comparative methods to data simulated under the coalescent framework in order to better understand the impact of discordance on phylogenetic inference. Our framework makes it possible to vary levels of ILS and the number of loci controlling a quantitative trait (cf. *Schraiber and Landis, 2015*), and so we also address whether variation in genetic architecture has an effect on phylogenetic inference. Finally, we use the threshold model (*Wright, 1934*; *Felsenstein, 2005*) to investigate whether discretizing quantitative traits makes inferences about them more or less robust to the potential effects of gene tree discordance and hemiplasy.

## Characterizing trait distributions in the three-species case under the coalescent and Brownian motion models

Before deriving results for the multispecies coalescent model using a three-species phylogeny, we present expectations for quantitative traits under Brownian motion (BM), a diffusion model commonly used in phylogenetic comparative methods. Although there are multiple possible interpretations of the underlying microevolutionary dynamics that lead to BM, we compare our model to it because it is tractable and popular, thus providing a clear baseline that is likely to be informative about the behavior of our model. Under BM, trait values from multiple species will exhibit a

multivariate normal distribution with the covariance structure given by the phylogeny (*Felsenstein, 1973*). More specifically, in the case of *n* species, the variances within species and covariances between species are given by $V = \sigma^2 T$, the variance-covariance matrix. Here, $\sigma^2$ is the evolutionary rate parameter, which measures how much change is expected in an infinitesimal time interval. $T$ is an *n* x *n* matrix whose off-diagonal entries, $t_{ij}$, are lengths of the internal branches subtending the ancestor of species *i* and *j*, and whose diagonal entries correspond to the lengths of the paths between each species and the root. For the phylogeny in *Figure 2a*, and $\sigma^2 = 1$:

$$V = \sigma^2 T = \sigma^2 \begin{bmatrix} t_{11} & t_{12} & t_{13} \\ t_{21} & t_{22} & t_{23} \\ t_{31} & t_{32} & t_{33} \end{bmatrix} = \begin{bmatrix} 5 & 4 & 0 \\ 4 & 5 & 0 \\ 0 & 0 & 5 \end{bmatrix} \tag{1}$$

For example, the BM expected variance in species A, $Var_{BM}(A)$, corresponds to the rate parameter multiplied by the length of the path extending from the root to the tip, and therefore evaluates to five. Note that $Var_{BM}(A)$ is not the population trait variance observed among individuals of A, but the expected variance in species A's mean trait value, resulting from evolution along the lineage leading to A. The covariance between species A and B, $Cov_{BM}(A, B)$, corresponds to the rate parameter multiplied by the length of the branch shared by these two lineages, which evaluates to four in this example.

Given the species tree topology in *Figure 2a*, the expected variance in trait value within any species, A, B, or C, is also readily derived under a neutral coalescent model (for a complete derivation, see section 1.1 in Appendix 1). Although many quantitative traits may be under selection, the selection coefficient on an individual allele is proportional its effect size (*Keightley and Hill, 1988*). This implies that loci with small effects will experience very small selection coefficients, and that neutral expectations for genealogical quantities should still be justified. The expected trait value variance within species A or B is given by:

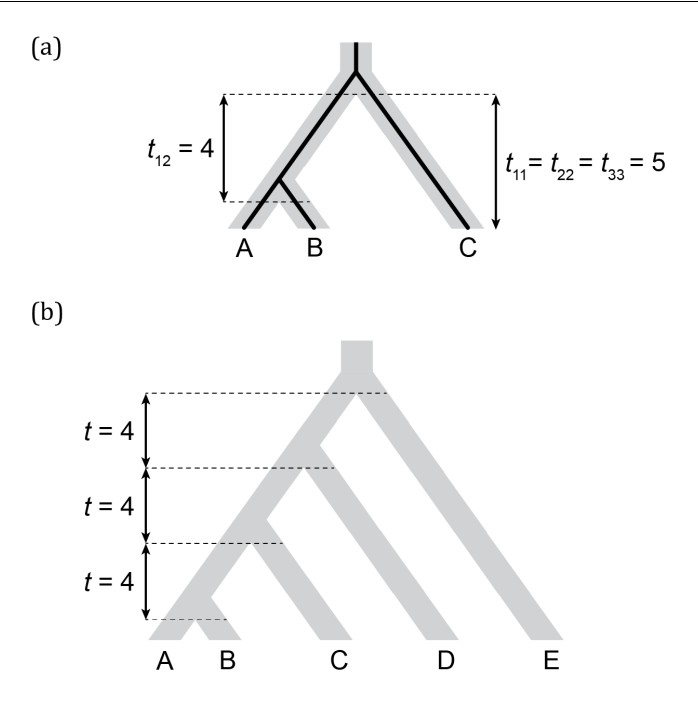

**Figure 2.** Three- and five-taxon trees with branch lengths indicated. (**a**) Three-species phylogeny (and a concordant gene tree within it) and its corresponding *T* matrix entries. (**b**) Five-species phylogeny used in coalescent simulations for PCM analyses. Branch lengths are indicated in units of 2*N* generations.
DOI: https://doi.org/10.7554/eLife.36482.003

$$Var_{Coal} = 2N\mu\sigma_M^2 \left[ t_e + \left(1 - e^{-t/2N}\right)\left(\frac{t}{2N} + 1\right) + \left(e^{-t/2N}\right)\left(\frac{t}{2N} + 1 + \frac{1}{3}\right) \right] \tag{2}$$

where $t$ is the length of the single internal branch of the species tree measured in generations, $t_e$ is the length of terminal branch leading to species A and B (i.e. the length of the A and B leaves), N is the population size, $\mu$ is the neutral mutation rate, and $\sigma_M^2$ is the variance of the mutational effect distribution. This last parameter describes the effect of individual mutations, and does not correspond to the Brownian motion evolutionary rate.

The term within square brackets in *Equation 2* captures the total amount of time the trait has been evolving since the common ancestor of all lineages in the tree, which is equivalent to the total length of root-to-tip paths in all gene trees underlying it. The first term inside the brackets ($t_e$) describes the length of the leaves, which are not affected by the coalescent process. The second term in the brackets computes the path length coming from gene trees in which the A and B lineages coalesce in their common ancestor (with probability $1 - e^{-t/2N}$), while the third term gives the path length coming from gene trees where A and B fail to coalesce in that ancestor (with probability $e^{-t/2N}$). These two kinds of gene trees contribute path lengths proportional to (t/2N + 1) and (t/2N + 1 + 1/3), respectively. Note that the expected trait value variance within species C is given by a sum similar to *Equation 2*, but that has the t/2N terms dropped from the path length component (and where $t_e$ corresponds to the terminal branch leading to species C; see section 1.1 in Appendix 1).

Following the same notation, the expected covariances in trait values between species are (for a complete derivation, see section 1.2 in Appendix 1):

$$Cov_{Coal}(A,B) = 2N\mu\sigma_M^2 \left[ \left(1 - e^{-t/2N}\right)\left(1 + \left(\frac{t}{2N} - \left(1 - \frac{t/2N}{e^{t/2N} - 1}\right)\right)\right) + \left(\frac{1}{3}e^{-t/2N}\right) \right] \tag{3}$$

and

$$Cov_{Coal}(A,C) = Cov_{Coal}(B,C) = 2N\mu\sigma_M^2 \left(\frac{1}{3}e^{-t/2N}\right) \tag{4}$$

Here, the covariances between species A and C and between B and C are the same because A and B are equally distant to species C. Note that the amount of independent evolution a species undergoes is irrelevant to the expected trait value covariances, which is why the $t_e$ term is missing from *Equations 3 and 4*.

With the expectations under BM and the coalescent in hand, we can now ask how these quantities compare in the simple case of little to no ILS (we use the species tree and branch lengths depicted in *Figure 2a*, for which the probability of discordance is very low, $\approx 0.02$). It is easy to see that for any $N > 0$, the single-species variance under the coalescent model will be larger than that expected under BM. For example, even in the extreme case where $2N = 1$ (and by setting $\sigma_M^2$ and $\mu$ = 1), we can observe that the variance within any of the species under the coalescent is higher ($Var_{Coal} = 6$) than under the BM model ($Var_{BM} = 5$). This is a curious, yet not unexpected result: traditional phylogenetic models such as BM do not consider the variation that exists in ancestral populations prior to speciation (*Gillespie and Langley, 1979*; *Edwards and Beerli, 2000*). Even though gene trees are always concordant with the species tree in this scenario, they will also always be taller due to the waiting times for coalescence in ancestral populations.

Conversely, expected covariances between species under both models should be exactly equivalent in the absence of genealogical discordance. First, the covariance between species A and C (and B and C) should be zero in both cases: these lineages do not share internal branches under either model when there is no ILS. Indeed, $Cov_{Coal}(A, C)$ and $Cov_{BM}(A, C)$ both evaluate to 0 in the absence of ILS, as specified by *Equation 4* and *Equation 1*, respectively. Second, the internal branch subtending species A and B is the same length in both models, as the waiting time for coalescence in the ancestral population of A and B is exactly the same as the waiting time in the ancestral population of all three species (*Figure 2a*). Therefore, $Cov_{Coal}(A, B)$ and $Cov_{BM}(A, B)$ both also evaluate to four in this scenario.

In summary, we can model the distribution of quantitative trait values across species under the coalescent model as a collection of contributions from many individual genealogies that all

determine the value of such a trait. However, expected trait values in the coalescent are not exactly the same as those expected under the classical BM model, even in the absence of genealogical discordance and given a fixed phylogeny. While expected covariances will be identical between models if no genealogical discordance is present, expected variances will still differ; this difference will be accentuated with larger ancestral population sizes. This result will therefore affect any parameters being estimated — such as the evolutionary rate, $\sigma^2$ — that depend on expected species variances. Below, we explore how the expectations under the coalescent and BM models can further differ in the presence of ILS and discordance.

## Consequences of genealogical discordance to quantitative traits: the 'deep coalescence' effect and hemiplasy

We can predict from the expectations laid out above that the variances and covariances under the coalescent model will change in the presence of discordance. In contrast, the BM model will have the same expectations because it does not consider genealogical discordance — the species tree is a fixed parameter. In order to characterize the effects of discordance on variances within species and covariances between species, we considered five different scenarios with increasing percentages of gene tree discordance (0, 15, 30, 45 and 60% discordance, respectively). We used the three-species phylogeny (*Figure 2a*) for its mathematical tractability, and in addition to computing the expectations of these measures (using *Equations 2–4*), we simulated 1000 data sets under each of the five scenarios. This simulation procedure is illustrated in *Figure 3*, where for each locus underlying a quantitative trait, mutations are thrown down at random along the genealogy and mutational effects of each mutation are drawn from a distribution determined by $\sigma_M^2$. Simulations were repeated for different numbers of loci affecting the trait: 5, 15, 25, 50 and 100. In keeping with the usual practice in

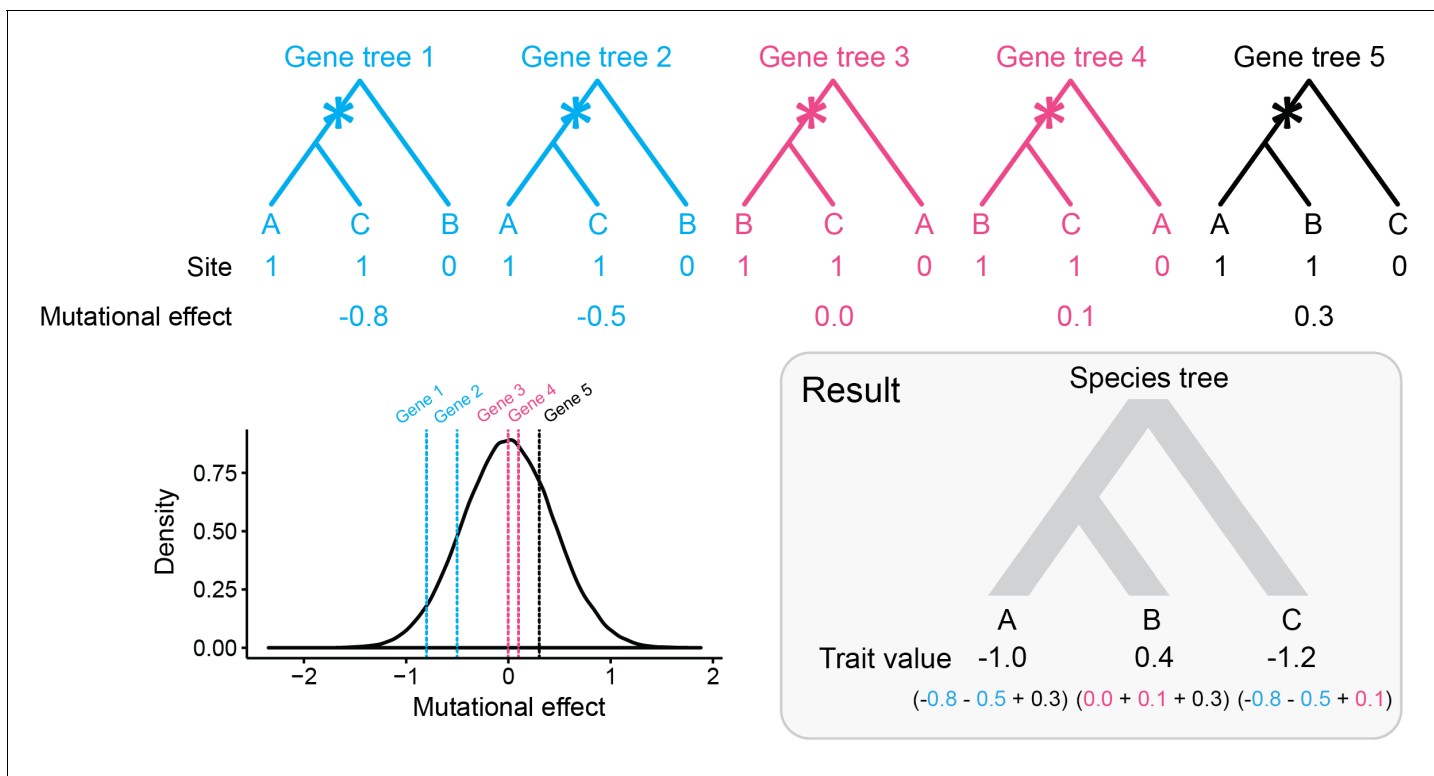

**Figure 3.** A single continuous traits controlled by five loci, four of which have discordant gene trees. (a) Genealogies of the loci controlling the trait. Asterisks represent mutations at a given site in each of the five loci. Ancestral alleles (0) have no effect on the trait value. Derived alleles (1) have their random effects on the trait value drawn from a mutational effect distribution (see (b)). (b) Mutational effect distribution of derived alleles. The distribution has a mean of zero and unit variance. (c) The outcome of a simulation consists of one trait value per species, which correspond to the sum of all derived allele mutational effects coming from all loci controlling the trait.
DOI: https://doi.org/10.7554/eLife.36482.004

comparative analyses of employing a single, static species tree, levels of ILS were increased by multiplying ancestral population sizes by incrementally larger factors — the topology and branch lengths of the species tree were kept constant (see Materials and methods for details).

We observed an overall good match between the observed and expected variances and covariances (*Figure 4* and *Figure 5a*). Under the coalescent model, larger ancestral population sizes make coalescent waiting times longer, and result not only in more ILS and more gene tree discordance, but also in taller trees on average. As expected (*Equations 2–4*), data sets simulated with larger *N* therefore had higher variances and covariances (*Figure 4a–b*). We refer to this phenomenon as the 'deep coalescence' (DC) effect. The DC effect occurs due to the increase in average gene tree height, relative to the species tree height, under the coalescent model with large population sizes (cf. *Gillespie and Langley, 1979*). We stress that (i) this effect is *not* due to discordance, and (ii) not only variances, but covariances among lineages that share a history in the species tree, are affected. The latter happens because, as mentioned above, it will take longer for any two lineages to coalesce given a larger population size (the parameter *N* controls this time in *Equations 3 and 4*). Consequently, the waiting time for the last coalescent event (which determines the length of the internal branch) will also be longer, leading to higher covariances between pairs of descendant species.

The number of loci did not influence variances and covariances (results not shown), which is expected. This is because the standard deviations of the mutational effect distributions used in our simulations ($\sigma_M^2$) are scaled to keep trait-value variances constant with changing numbers of loci, thus ensuring a fair comparison between models with different numbers of loci. This follows the standard logic of the infinitesimal model, that is, the larger the number of loci controlling a trait, the smaller the effect each mutation should have on the trait value (*Fisher, 1919*; for more details in the context of the coalescent model, see *Schraiber and Landis, 2015*).

Finally, under the coalescent model, gene tree discordance *does* have an effect: the covariance between species *A* and *C* (and between *B* and *C*) increases with more ILS relative to the covariance between species *A* and *B* (*Figure 5a*). Recall that when there is no discordance there is no covariance between non-sister species, because they do not share an evolutionary history. Discordant gene trees offer the opportunity for non-sister species to have a shared history, and covariance increases. As a result, there is an increased similarity between non-sister species in quantitative traits due to hemiplasy in the underlying gene trees. Ultimately, the effect of hemiplasy on continuous traits is to make covariances between different pairs of species converge on the same value (*Figure 5b*). This makes intuitive sense, as in the limit all three topologies will be equally frequent, resulting in equal covariances between all pairs of species.

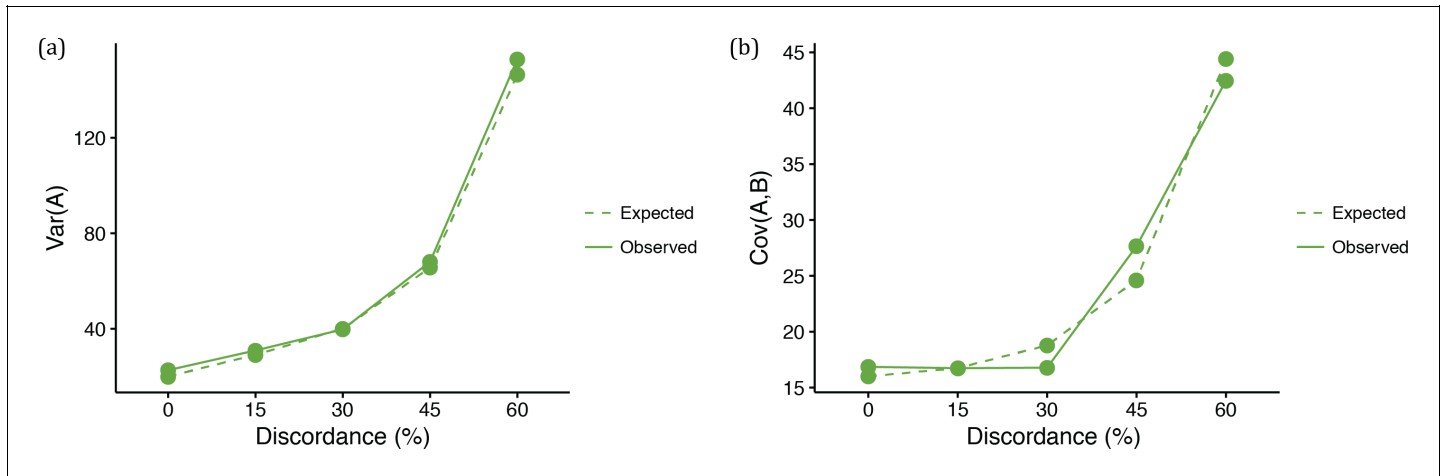

**Figure 4.** Quantitative trait summaries under varying levels of genealogical discordance, for a three-taxon species tree. (a) Expected and observed variances in trait values of species *A* in each of the five ILS conditions. Expected values come from *Equation 2*. (b) Expected and observed covariances between species *A* and *B* in each of the five ILS conditions. Expected values come from *Equation 3*.
DOI: https://doi.org/10.7554/eLife.36482.005

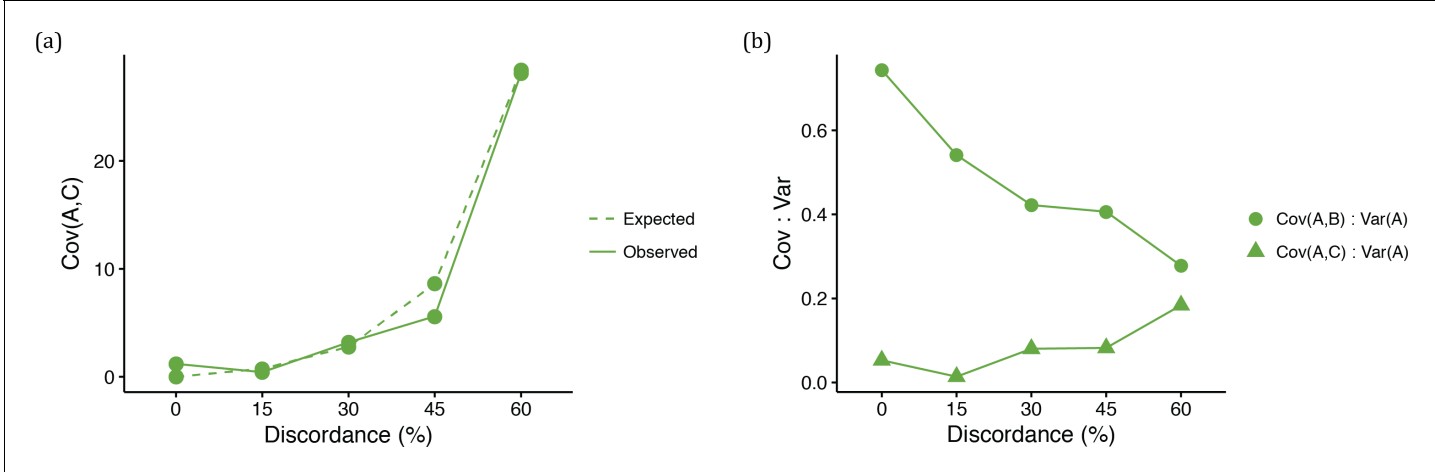

**Figure 5.** Quantitative trait summaries under varying levels of genealogical discordance, indicating the effect of hemiplasy for a three-taxon species tree. (a) Expected and observed covariances between species *A* and *C* in each of the five ILS conditions. Expected values come from *Equation 4*. (b) Observed covariances between a pair of species normalized by the variance in species *A*, for all five ILS conditions.
DOI: https://doi.org/10.7554/eLife.36482.006

We emphasize that the aforementioned effects were observed despite the fact that the concordant topology was always the most common, and that substitutions on discordant trees and discordant branches occurred in only a fraction of the loci underlying the continuous trait. Furthermore, the number of loci does not seem to strongly affect our results (see below), as the difference between covariances were similar regardless of the number of loci controlling the trait. This suggests that our model applies to all quantitative traits, not just those controlled by a large number of loci.

## Hemiplasy increases inferred evolutionary rates and decreases phylogenetic signal

The results from our model demonstrate that in the presence of ILS, quantitative traits will have both larger variances in each species as well as positive covariances between non-sister species. These outcomes suggest that hemiplasy could affect inferences involving quantitative traits, as traditional models (BM or otherwise) do not allow for shared trait variation along branches that do not exist in the species tree.

We first investigated the impact of discordance and hemiplasy on estimates of a commonly inferred parameter, the BM evolutionary rate, $\sigma^2$. We estimated $\sigma^2$ from data simulated along a five-species asymmetric phylogeny (*Figure 2b*). Simulating data for five species allows for more ILS (and a greater effect of hemiplasy; *Mendes and Hahn, 2016*) relative to the three-species case, due to the larger number of possible gene tree topologies (105 in the former case versus the three possible topologies in the latter). Again, we simulated data under five ILS conditions with different percentages of gene tree discordance (0, 20, 40, 60 and 80% discordant trees, respectively) by keeping the phylogeny constant and increasing population sizes. As in the three-species case, we simulated continuous traits controlled by 5, 15, 25, 50 and 100 loci.

As mentioned above, increasing ancestral population sizes increases both ILS and the average height of gene trees with two main resulting patterns: (i) expected covariances between non-sister species will increase (due to hemiplasy), and (ii) expected variances within species will increase (due to deep coalescence). Because BM does not model the number, topology, or lengths of the gene trees underlying a continuous trait, we predicted that both outcomes would be accounted for when inferring rates under the BM model as spuriously higher evolutionary rates. Indeed, we observed a positive correlation between the estimated $\sigma^2$ under BM and ILS (*Figure 6a*). This pattern was the same for all data sets, irrespective of the number of loci controlling the trait.

We also reasoned that another major consequence of hemiplasy — resulting from the changes in expected covariances in trait values between pair of species — would be the reduction of the average phylogenetic signal with increasing ILS. This is because the effect of hemiplasy on quantitative

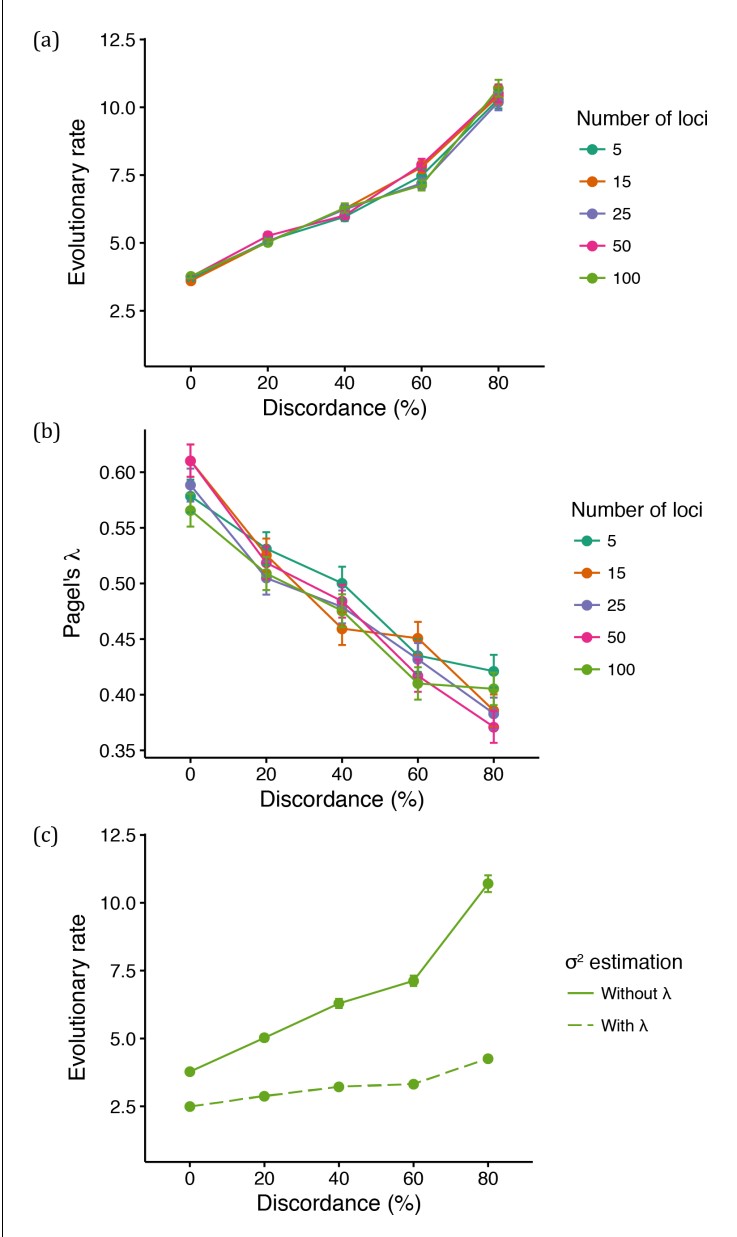

**Figure 6.** Mean parameter estimates under varying levels of genealogical discordance and number of loci controlling quantitative trait. (**a**) Mean evolutionary rate for different number of loci controlling the simulated continuous trait and different levels of discordance. (**b**) Mean value of Pagel's λ for different number of loci controlling the simulated continuous trait and different levels of discordance. (**c**) Mean evolutionary rate when 100 loci control the trait ('Without λ' is the same as shown in (**a**); in 'With λ', the rate was estimated with Pagel's λ).
DOI: https://doi.org/10.7554/eLife.36482.007

traits is to make covariances between more closely related species become smaller relative to covariances between more distantly related species. The more hemiplasy, the less should the covariances resemble values that would be observed for a trait evolving along the species tree, and thus the phylogenetic signal should be lower. We measured the phylogenetic signal in each replicated simulation by estimating a commonly used parameter, Pagel's λ (where λ = 1 indicates a trait evolving according to BM along a species tree, and λ <1 indicates lower phylogenetic signal; *Pagel, 1999*; *Freckleton et al., 2002*). As expected, estimates of λ decreased on average with increasing ILS

(*Figure 6b*), reflecting the lower phylogenetic signal of traits partly determined by discordant gene trees.

Given the results from Pagel's λ, we attempted to distinguish the contribution of the DC effect (i. e. overall increase in variances and covariances) from that of hemiplasy (i.e. relative change in covariances) to the spurious increase in $\sigma^2$. The parameter λ can be thought of as a species tree branch-stretching parameter (*O'Meara, 2012*): we predicted that when estimating $\sigma^2$ in the presence of λ, the latter would act as a 'buffer' parameter absorbing the effect of hemiplasy by becoming reduced itself (as shown in *Figure 6b*). Indeed, evolutionary rates were much lower when estimated in the presence of λ (*Figure 6c*, 'With λ'), but still remain higher in data sets simulated with increasing levels of ILS. This is because while λ can absorb the effect of hemiplasy by shrinking internal branches, it cannot account for the DC effect resulting from the increased average gene tree heights in higher ILS conditions.

These results suggest that both the DC effect and hemiplasy contribute to the increase in estimates of $\sigma^2$. In BM model terms, understanding the impact of the DC effect on higher estimates of $\sigma^2$ is straightforward: if the tree (reflected in matrix $T$, *Equation 1*) is held constant and all variances and covariances (the entries of $V$, *Equation 1*) become larger, then $\sigma^2$ must become larger. But our results also suggest that the effect of hemiplasy is comparable to the DC effect, and possibly of even greater magnitude in the presence of more ILS. This observation is perhaps less intuitive, but indicates that $\sigma^2$ must become much higher to account for the difference between the observed covariances (i.e. off-diagonal entries of $V$) and expected covariances, given the observed variances and $T$. Assuming that quantitative traits evolve according to the coalescent model, larger ancestral population sizes and genealogical discordance can thus lead to an overestimation of $\sigma^2$ and to lower values of λ, and will likely affect all comparative methods that can make use of such parameters, not just the BM model. We point the curious reader to Appendix 1 (section 2.3) for a thorough theoretical treatment on how expected trait variances and covariances under the two models should differ, and why these differences can lead to the reported estimates of $\sigma^2$ and λ.

## Hemiplasy can increase the false positive rate in phylogenetic hypothesis testing

Many studies test the hypothesis that groups of species differ in measured traits due to factors other than phylogenetic relatedness. We addressed whether hemiplasy could also interfere with this type of phylogenetic hypothesis testing. The comparative method of choice we used was the phylogenetic ANOVA (*Garland et al., 1993*). As in traditional ANOVA, this method allows the comparison of mean trait values across groups of species. Within a linear model framework, the phylogenetic ANOVA also corrects for the inflation of degrees of freedom caused by the non-independence of trait value errors around the regression line (which can be estimated by looking at the residuals) — which results from the hierarchical nature of the phylogenetic relationships among species (*Felsenstein, 1985*; *Garland et al., 1993*; *Uyeda et al., 2018*). This correction allows the approximation of the true number of degrees of freedom through simulations of trait values along the phylogeny (given some model of trait evolution — BM in our case). The simulations collectively comprise an empirical $F$ distribution that is then used in hypothesis testing (*Garland et al., 1993*).

Our prediction was that increasing levels of ILS and of hemiplasy would increase the false positive rate of phylogenetic ANOVAs. We tested this prediction by conducting phylogenetic ANOVAs on the five-species simulations. Hypothesis testing consisted of comparing the null hypothesis that a pair of species had the same mean trait value as the remaining three species, against the alternative hypothesis of different means. This procedure was repeated on each of the 1000 replicates, for all possible groupings of two species versus three species; we then recorded the average number of times per replicate the p value was significant ($p < 0.05$).

As predicted, we observed a positive correlation between ILS levels and the mean number of times the null hypothesis was rejected in the phylogenetic ANOVA; this trend was unaffected by the number of loci underlying the trait (*Figure 7*). This result suggests that an arbitrary group of species is, on average, more likely to have a spuriously (and significantly) smaller or larger mean trait value than the remaining species in the presence of gene tree discordance.

The sharing of similar trait values by non-sister species is an expected byproduct of higher expected covariances among those species when there is gene tree discordance. Such changes in

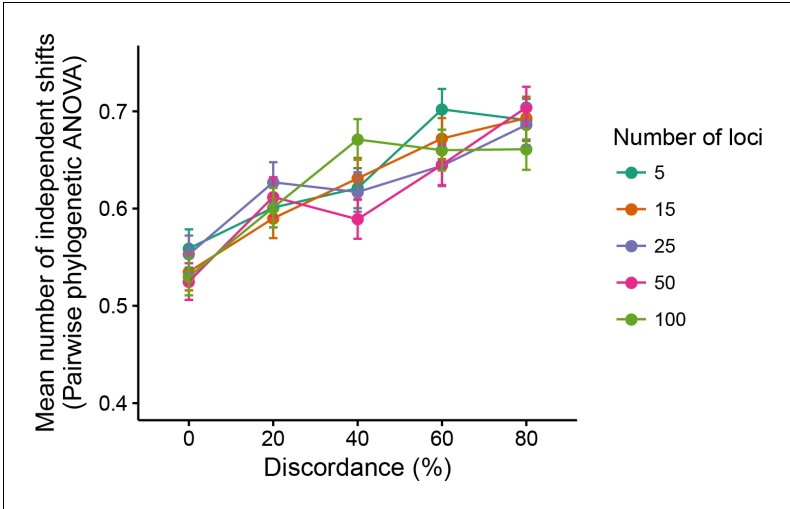

**Figure 7.** Mean number of independent trait-value shifts (i.e. significant phylogenetic ANOVA tests) among all possible groupings of two versus three species.
DOI: https://doi.org/10.7554/eLife.36482.008

expected covariances between pairs of species trait values are a symptom of hemiplasy (*Figure 5*), but not of the DC effect. We thus believe that hemiplasy not only contributes to the incorrect estimation of parameters such as the evolutionary rate, but can also play a role in increasing the false positive rate of phylogenetic comparative methods.

## Threshold traits are strongly affected by hemiplasy

As demonstrated above, the magnitude of the effect of hemiplasy on phylogenetic inferences is consistently proportional to the observed levels of gene tree discordance in a data set. Our results also suggest that the number loci underlying a quantitative trait does not matter to the expected trends from such inferences. One remaining question, however, is whether hemiplasy can have an effect on a threshold trait — that is, a discrete trait that has a continuous character underlying it (*Wright, 1934*; *Felsenstein, 2005*), and if the genetic architecture of such trait is relevant to this effect.

Addressing this question is straightforward, as we only need to treat our simulated continuous traits as the underlying 'liability' of a threshold character. By choosing an arbitrary threshold of one standard deviation above the mean continuous trait value (over all replicates and all species), we coded all simulated trait values as either '0' (if below the threshold), or '1' (if above). Defining a threshold using a dispersion measure such as the standard deviation, instead of a fixed value, allows us to account for the higher variances expected in replicates under higher ILS conditions.

Before laying out our predictions for how the effect of hemiplasy on threshold characters should be manifested, we first define a few terms used in the discussion that follows. A 'trait pattern' consists of the threshold character states (from a single replicate) observed at the tips of the tree. Given tree ((((A,B),C),D),E) (the tree we used in the simulations; *Figure 2b*), trait pattern '11000' signifies species *A* and *B* sharing state '1' (both had liabilities above the threshold) and species *C*, *D* and *E* sharing state '0' (the three species had liabilities below the threshold). A congruent (informative) trait pattern can be produced by character-state transitions occurring on internal branches that are present in the species tree; thus trait patterns '11000' and '11100' are congruent. Conversely, an incongruent trait pattern is the result of either homoplastic or hemiplastic evolution: multiple true character-state transitions, or transitions on internal branches of discordant gene trees that are absent from the species tree, respectively. Trait patterns such as '01100' and '11010', for example, are incongruent.

If hemiplasy affects threshold traits as it does continuous traits, we predicted that higher ILS levels would lead to a larger number of incongruent trait patterns, and to a lower number of congruent trait patterns. As expected, counts of incongruent informative trait patterns increased with

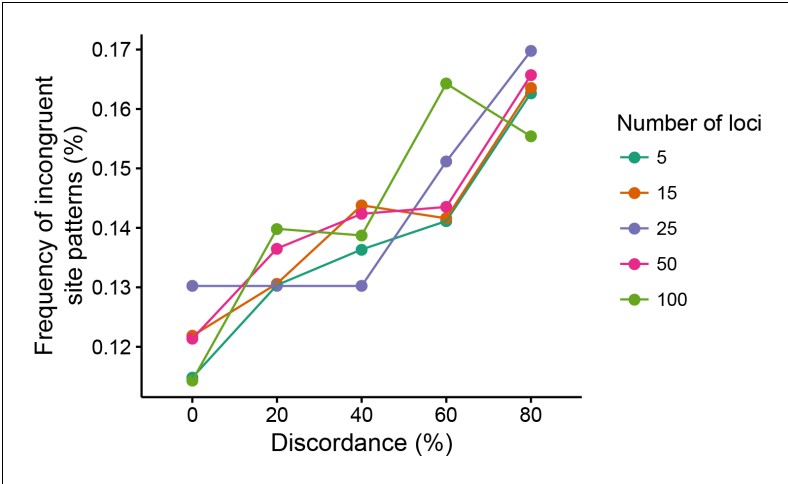

**Figure 8.** Frequency of incongruent trait patterns (out of all informative trait patterns) for threshold traits. Each combination of level of discordance and number of loci was simulated 1000 times.
DOI: https://doi.org/10.7554/eLife.36482.009

increasing ILS levels (*Figure 8*); congruent trait patterns likewise decreased. Furthermore, the same trend was observed from simulations where the liability character was underlain by few or many loci (*Figure 8*). This suggests that even in the case of threshold traits, larger numbers of loci comprising the genetic architecture will not mitigate the effect of hemiplasy.

Incongruent trait patterns are interesting because they can be suggestive of convergent evolution, or of correlated evolution when more than one trait exhibits similar patterns. While we do not further investigate the behavior of phylogenetic comparative methods applicable to discrete characters here, it is clear nonetheless that in the presence of gene tree discordance inferences from incongruent patterns can be misleading about the number of times a trait has evolved. This is because misidentification of the branches along which character states are inferred to change will be more likely in the presence of gene tree discordance. Overall, the more genealogical discordance is present in a data set, the more likely it is that a discrete trait will exhibit an incongruent pattern by chance, simply as a result of the stochasticity of the coalescent process.

## Discussion

In the present study, we propose a multispecies coalescent model for quantitative trait evolution that incorporates the genealogical discordance that underlies such traits. We use this model to investigate whether discordance can affect phylogenetic inferences, mainly through the phenomenon of hemiplasy. We considered ILS to be the sole cause of gene tree discordance. By employing coalescent theory, we demonstrate that in the absence of ILS the coalescent and traditional BM model are equivalent with respect to the expected covariances between species trait values, but differ in terms of the species expected trait variances. In the presence of ILS, hemiplasy causes the expected covariance in trait values between pairs of more distantly related species to increase.

The increased covariance due to hemiplasy leads to error in estimates of two parameters commonly studied under the BM model, namely, the evolutionary rate, $\sigma^2$, and Pagel's $\lambda$. Hemiplasy consistently led to an overestimation of $\sigma^2$, and to lower $\lambda$ estimates. Moreover, errors were also observed when conducting comparative analyses such as the phylogenetic ANOVA, whose false positive rate was increased by greater levels of genealogical discordance and hemiplasy. Finally, by treating quantitative traits as a liability character underlying a threshold trait, we found that hemiplasy affects the number of times such traits appeared incongruent with the species tree. All of the aforementioned results held irrespective of the number of loci controlling the quantitative trait.

We have focused the comparisons of trait-value expectations from our model (which assumes no selection) to those under BM. The latter can be seen as equivalent to a quantitative genetics model in which many genes have small effects on a selectively unconstrained character (*Lande, 1976*;

*Harmon, 2017*). However, evidence suggests that even when BM fits the data well, the evolutionary rate parameter does not seem to be meaningful as a quantitative genetic measure of drift (*Estes and Arnold, 2007*; *Harmon et al., 2010*; *Uyeda et al., 2011*). We note again that the motivation for investigating BM (and methods that employ BM, e.g. the phylogenetic ANOVA) as a model for learning about quantitative traits was not the similarity of its assumptions to the theoretical and simulation conditions considered here, but instead: (i) its popularity, tractability, and centrality in these types of studies, and (ii) the fact that BM-based models do not explicitly incorporate the number of loci, their gene trees, nor the effects of each locus on a quantitative trait of interest. These qualities of BM are helpful in elucidating the properties of our model, in addition to being suggestive of how more realistic BM-based models would fare in the face of discordance.

There are extensions to BM that can explicitly include certain forms of selection (e.g. the Ornstein-Uhlenbeck model; *Butler and King, 2004*) but whose behavior we have not investigated. While these models incorporate selection, it is still unlikely that the parameters of such models can be interpreted as population-genetic quantities. For the purposes of the analyses presented here, whether or not BM can accommodate traits evolving under a mix of drift and selection is relevant only insofar as this model can escape the effects of hemiplasy. Based on our results, we expect that models that do not explicitly consider genealogical discordance can at least in principle be vulnerable to hemiplasy, regardless of whether they include selection.

Although we have only presented theoretical results for trees with a small number of tips, our results are likely to hold even when the species tree is large. This is because the total number of taxa is not especially relevant to the multispecies coalescent process; instead, it is the number of lineages in each phylogenetic 'knot' (i.e. the number of lineages involved in ILS; *Ané et al., 2007*). Extensions of our results to knots with four or five taxa are straightforward, though with larger numbers than this the total number of topologies that need to be considered grows unwieldy. Similarly, the assumptions used by our model about constant population sizes and mutation rates are only required to be true across phylogenetic knots, where the relevant coalescent events occur. This may be only a very small fraction of the total amount of time represented by large trees. We do not expect violations of these assumptions elsewhere in the tree — outside of knots — to have a qualitative effect any different than they would in non-coalescent models. Of course, the larger the periods of time covered by any such analysis, the harder it can be to infer parameters of interest, as in those circumstances even estimating phylogenies can be a daunting task. In such cases, ILS may be the least of a phylogeneticist's concerns. Finally, when analyzing our simulated data here we employed the true, known species tree used in the simulations, but in typical phylogenetic analyses of real data an estimate of the phylogeny is used. These facts will certainly cause results from real-world studies to deviate from expectations.

While we found that discordance can indeed affect traditional phylogenetic methods for studying traits with complex genetic architectures, it is likely that the results from our simple simulations are conservative. This is because we assumed only additivity of mutations and a Gaussian mutational effect distribution. The presence of dominance, epistasis, and broader or skewed mutational effect distributions are likely to compound the effects of hemiplasy. Moreover, while our assessment of phylogenetic methods is by no means exhaustive, it is unlikely that the trends we report here are exclusive to the approaches we investigated. Methods that compare models with one versus multiple evolutionary rates across a tree (*O'Meara et al., 2006*), or that estimate branch lengths from quantitative traits (*Felsenstein, 1973*), for example, could be affected by hemiplasy in the same way that nucleotide substitution models are (*Mendes and Hahn, 2016*). Similarly, methods addressing the correlation between discrete traits (e.g. *Pagel, 1994*) could also have increased false positive rates if hemiplasy acts on multiple traits in similar ways. Hemiplasy is also expected to broaden the confidence intervals around ancestral state reconstructions of quantitative traits (*Martins and Hansen, 1997*), making it harder to infer significant shifts in trait means and to place such shifts on internal branches of the species tree. While recently proposed methods that study BM models over species networks do represent a step forward in the presence of discordance due to hybridization and introgression (e.g. *Dwueng-Chwuan and O'Meara, 2015*; *Bastide et al., 2018*), these methods do not account for either deep coalescence or the full spectrum of genealogical discordance.

Given our results, it is reasonable to ask whether and when traditional phylogenetic comparative methods for quantitative traits are appropriate. ILS is expected to act when there are short internode distances in a species tree, regardless of how far in the past the rapid succession of speciation

events has occurred. This implies that the effects of hemiplasy will be greatest for species radiations, as these are defined as periods of rapid speciation (*Schluter et al., 1997*; *Freckleton and Harvey, 2006*). Conversely, many phylogenetic studies include species trees without much discordance, either because none has occurred or because taxa have been chosen to minimize discordance. This latter approach — thinning species from analyses in order to minimize or remove discordance — can improve the accuracy of inferences about molecular changes (*Mendes and Hahn, 2016*). Another recommendation for avoiding the problems associated with discordance would be to use Pagel's λ in analyses aimed at estimating evolutionary rates under the BM model. We have shown here that Pagel's λ will 'buffer' the effect of hemiplasy, minimizing the overall impact of genealogical discordance over the estimates.

Good models are able to strike a balance between biological realism and tractability. While the coalescent model has the potential of being more realistic than the family of models based on BM, the inferential machinery for the coalescent is less well-developed. It is thus still unclear how tractable the coalescent model would be in terms of phylogenetic inference. On the other hand, numerous comparative methods making use of BM's simplicity and tractability (e.g. the existence of analytical solutions for maximizing the likelihood of parameters of interest) have been developed, implemented, and tested over the years, and so we do not expect a complete shift away from such methods in the near future. Moving forward, we nonetheless believe it worthwhile to further develop and explore models with the potential of being more realistic and more robust to problems such as those described here. At the least, phylogenetic analyses using the BM model on trees with underlying discordance must be examined carefully.

## Materials and methods

### Simulations under the multi-species coalescent model for quantitative traits

In order to simulate a quantitative trait evolving along a species tree, we extended the population model put forward in *Schraiber and Landis, 2015*) into a phylogenetic model (*Figure 3*), and modified the tools made available by these authors accordingly. As in traditional phylogenetic models, the trait value simulated for a species was treated as the mean of its populations (*Harmon, 2017*). Each species trait value corresponded to the sum of the effects of derived alleles (ancestral alleles had no effect on trait values) at variant sites from all loci controlling the trait. The effect of each derived allele was drawn from a normal mutational effect distribution with mean zero and standard deviation scaled by the number of loci underlying the trait (e.g. *Figure 3b*).

Simulations were conducted with the coalescent simulator *ms* (*Hudson, 2002*) along species trees ((*A*:1,*B*:1):4,*C*:5) and ((((*A*:1,*B*:1):4,*C*:5):4,*D*:9):4,*E*:13). We simulated traits underlain by varying numbers of loci (5, 15, 25, 50 and 100), each under five ILS conditions with increasing amounts of gene tree discordance (1000 replicates per condition). Gene tree discordance was introduced by simulating larger ancestral populations; we did so by multiplying the size of these populations by an increasingly larger factor while keeping species tree branch lengths constant. In the three-species phylogeny case, the five ILS conditions differed by increments of 15% in gene tree discordance (where *N* was multiplied by factors of 1, 5.2, 9.6, 19.5 and 50), with the lowest and highest ILS conditions exhibiting 0 and 60% discordance, respectively. In the five-species phylogeny case, increments in gene tree discordance were of 20% (factors were 1, 3.6, 5.6, 8 and 14), with 0 and 80% of gene trees being discordant in the lowest and highest ILS condition, respectively. We fixed $\theta$ = 4 in all simulations.

### Parameter estimation and hypothesis testing under Brownian motion

Parameter estimation was carried out for each of the 5000 replicated simulations along the five-species phylogeny (1000 per ILS condition). We estimated the evolutionary rate, $\sigma^2$, and λ parameters with the 'fitContinuous' function of R's *geiger* package (*Harmon et al., 2008*). We further inferred $\sigma^2$ in the presence of λ ('With λ' in *Figure 6c*) using the same function in *geiger*.

Phylogenetic ANOVA was also conducted on all simulations from the five-species phylogeny to test the hypothesis that a pair of species shared the same mean trait value, while the other three species had a different mean. We performed one test for each possible pair of species (versus the

remaining three species), and repeated these tests for each replicated simulation under all ILS conditions. We then counted for each replicate how many of these tests yielded a significant p value ($p < 0.05$). Again, phylogenetic ANOVA was carried out with the *geiger* package.

## Discretization of quantitative trait values using the threshold model

In order to characterize the effect of hemiplasy on threshold traits, we treated the quantitative trait simulated with the five-species phylogeny as a continuous liability (*Wright, 1934*; *Felsenstein, 2005*). Each species liability value was then compared to a threshold to generate the species' corresponding trait value: '0' if below the threshold, and '1' if above. Exact threshold values were adjusted between ILS conditions to account for the fact that data sets with more gene tree discordance should have greater expected variances in liability values. We set the threshold of a given ILS condition to the value at one standard deviation above the mean of the liability value distribution (from all species and all replicates) for that ILS condition. ILS conditions with more gene tree discordance thus had higher threshold values. We then tabulated all different informative trait patterns in which two species shared state '1', and the remaining three species shared state '0', and classified them as either congruent or incongruent (see main text).

## Acknowledgements

The authors would like to thank Matt Pennell and two other reviewers for comments that improved this article. FKM and MWH were supported by National Science Foundation grant DBI-1564611. JGS was supported by National Institutes of Health grant R35 GM124745.

## Additional information

### Funding

| Funder | Grant reference number | Author |
|---|---|---|
| National Science Foundation | DBI-1564611 | Matthew W Hahn<br>Fábio K Mendes |
| National Institutes of Health | R35 GM124745 | Joshua G Schraiber |

The funders had no role in study design, data collection and interpretation, or the decision to submit the work for publication.

### Author contributions

Fábio K Mendes, Conceptualization, Formal analysis, Writing—original draft, Writing—review and editing; Jesualdo A Fuentes-González, Joshua G Schraiber, Formal analysis; Matthew W Hahn, Conceptualization, Writing—original draft, Writing—review and editing

### Author ORCIDs

Fábio K Mendes http://orcid.org/0000-0001-6204-7208
Matthew W Hahn https://orcid.org/0000-0002-5731-8808

### Decision letter and Author response

Decision letter https://doi.org/10.7554/eLife.36482.017
Author response https://doi.org/10.7554/eLife.36482.018

## Additional files

### Supplementary files

• Source code file 1. R script for plotting figures.
DOI: https://doi.org/10.7554/eLife.36482.010

• Transparent reporting form
DOI: https://doi.org/10.7554/eLife.36482.011

## Data availability

Simulated data can be found at: https://doi.org/10.5061/dryad.m2s1735

The following dataset was generated:

| Author(s) | Year | Dataset title | Dataset URL | Database, license, and accessibility information |
|---|---|---|---|---|
| Mendes FK, Fuentes-González JA, Schraiber JG, Hahn MW | 2018 | Data from: A multispecies coalescent model for quantitative traits | https://doi.org/10.5061/dryad.m2s1735 | Available at Dryad Digital Repository under a CC0 Public Domain Dedication |

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

## Appendix 1

DOI: https://doi.org/10.7554/eLife.36482.012

# 1. Deriving expected variances and covariances in quantitative trait values under the coalescent model

### 1.1 Variance in the three species case

Given a three-species phylogeny $S$, the variance in values of trait $X$ within a diploid species $i$ is defined as:

$$\mathrm{Var}(X_i) = 2N\mu\sigma_M^2 \times B_{\mathrm{root},i}, \tag{S.1}$$

where $N$ is the constant population size along the phylogeny, $\mu$ the mutation rate, and $\sigma_M^2$ the variance of the mutational distribution. (Note that $\sigma_M^2$ is not the Brownian motion evolutionary rate, $\sigma^2$, which is instead equivalent to $\mu\sigma_M^2$.) $B_{\mathrm{root},i}$ is the expected total length of all branch paths from the root to species $i$ coming from all *gene trees* generated by $S$.

We can further expand $B_{\mathrm{root},i}$ as:

$$B_{\mathrm{root},i} = t_e + \left(1 - e^{-t/2N}\right)\left(\frac{t}{2N} + 1\right) + \left(e^{-t/2N}\right)\left(\frac{t}{2N} + 1 + \frac{1}{3}\right), \tag{S.2}$$

where $t$ and $t_e$ are internal and terminal branch lengths from $S$ in generations, $1 - e^{-t/2N}$ is the probability that the sister lineages coalesce in their ancestor (i.e. a concordant gene tree is observed), and $e^{-t/2N}$ the probability that they do not (i.e. the three lineages enter their MRCA and then a concordant or discordant gene tree can be observed). These probabilities then multiply the contributions of their corresponding gene trees to the total path length. Concordant gene trees whose lineages sort in their immediate ancestor contribute a path length of $t_e + \frac{t}{2N} + 1$, while all other trees (concordant or discordant) contribute $t_e + \frac{t}{2N} + 1 + \frac{1}{3}$.

We can now arrive at **Equation 2** from the main text:

$$E(\mathrm{Var}(X_i)) = 2N\mu\sigma_M^2\left[t_e + \left(1 - e^{-t/2N}\right)\left(\frac{t}{2N} + 1\right) + \left(e^{-t/2N}\right)\left(\frac{t}{2N} + 1 + \frac{1}{3}\right)\right]. \tag{S.3}$$

## 2. Covariances in the three species case

Following the same notation, the covariance between trait values of species $i$ and $j$ is:

$$\mathrm{Cov}\left(X_i, X_j\right) = 2N\mu\sigma_M^2 \times B_{\mathrm{root},\mathrm{MRCA}_{(i,j)}}, \tag{S.4}$$

where $B_{\mathrm{root},\mathrm{MRCA}_{(i,j)}}$ is the expected total length of all gene tree branch paths from the root to the most recent common ancestor of species $i$ and $j$.

Given $S = ((A, B), C)$, and if we let $i = A$ and $j = B$, we can expand $B_{\mathrm{root},\mathrm{MRCA}_{(i,j)}}$ into:

$$B_{\mathrm{root},\mathrm{MRCA}_{(A,B)}} = \left(1 - e^{-t/2N}\right)\left(1 + \left(\frac{t}{2N} - \left(1 - \frac{t/2N}{e^{t/2N} - 1}\right)\right)\right) + \left(\frac{1}{3}e^{-t/2N} \times 1\right), \tag{S.5}$$

where term $e^{-t/2N}$ is multiplied by $\frac{1}{3}$ because species A and B are sister taxa in only one of the three possible equifrequent topologies. Again, concordant gene trees in which the A and B lineages coalesce in their most recent common ancestor occur at frequency $1 - e^{-t/2N}$, but we must now subtract the waiting time for their coalescence from their branch length contribution to $B_{\mathrm{root},\mathrm{MRCA}_{(A,B)}}$. This waiting time is given by $1 - \frac{t/2N}{e^{t/2N} - 1}$ and has been derived elsewhere (**Mendes and Hahn, 2018**). Note that concordant gene trees in which both coalescent events happen in the MRCA of the three species contribute to $B_{\mathrm{root},\mathrm{MRCA}_{(A,B)}}$ with a branch that is $2N$

generations long (the expected time to coalescence of two lineages), so $\frac{1}{3}e^{-t/2N}$ is multiplied by $2N \times \frac{t}{2N} = 1$.

We now arrive at **Equation 3** from the main text:

$$E(\mathrm{Cov}(X_A, X_B)) = 2N\mu\sigma_M^2 \left[ \left(1 - e^{-t/2N}\right) \left(1 + \left(\frac{t}{2N} - \left(1 - \frac{t/2N}{e^{t/2N} - 1}\right)\right)\right) + \left(\frac{1}{3}e^{-t/2N}\right) \right]. \quad \text{(S.6)}$$

Finally, if we let $i = \mathrm{A}$ (or $i = \mathrm{B}$) and $j = \mathrm{C}$, $B_{\mathrm{root,MRCA}_{(i,j)}}$ is defined as:

$$B_{\mathrm{root,MRCA}_{(A,C)}} = B_{\mathrm{root,MRCA}_{(B,C)}} = \frac{1}{3}e^{-t/2N} \times 1. \quad \text{(S.7)}$$

As in **Equation S.5**, each discordant gene tree contributes with a branch that is $2N$ generations long, and so its probability $\frac{1}{3}e^{-t/2N}$ is multiplied by $2N \times \frac{1}{2N} = 1$. From this equation, we can then derive **Equation 4** from the main text:

$$E(\mathrm{Cov}(X_A, X_C)) = E(\mathrm{Cov}(X_B, X_C)) = 2N\mu\sigma_M^2 \left(\frac{1}{3}e^{-t/2N}\right). \quad \text{(S.8)}$$

## 2. An alternative derivation for variances and covariances in quantitative traits, with further considerations

As in the previous sections, we embed our derivations in a phylogenetic context by assuming that we have $n$ tips in a species tree and exactly one sample per species. For the sake of simplicity in terms of notation, we measure branch lengths in generations instead of units of $2N$ generations (as above and in the main text); this accounts for missing factors of $2N$ in all equations below relative to equations in the main text and above. Given a trait controlled by $L$ independent loci, and letting $X_{i,l}$ be the contribution of locus $l$ to $X_i$ (the value of trait $X$ in species $i$) we assume an additive model, that is that

$$X_i = \sum_{l=1}^{L} X_{i,l}. \quad \text{(S.9)}$$

Because the loci are independent and identically distributed, the variance and covariance of the trait can be computed by summing over loci,

$$\begin{aligned} \mathrm{Var}(X_i) &= \sum_{l=1}^{L} \mathrm{Var}(X_{i,l}) \\ &= L\mathrm{Var}(X_{i,l}) \end{aligned} \quad \text{(S.10)}$$

and

$$\begin{aligned} \mathrm{Cov}(X_i, X_j) &= \sum_{l=1}^{L} \sum_{k=1}^{L} \mathrm{Cov}(X_{i,l}, X_{j,k}) \\ &= \sum_{l=1}^{L} \mathrm{Cov}(X_{i,l}, X_{j,l}) \\ &= L\mathrm{Cov}(X_{i,l}, X_{j,l}) \end{aligned} \quad \text{(S.11)}$$

where the second line follows because $\mathrm{Cov}(X_{i,l}, X_{j,k}) = 0$ if $k \neq l$ by assumption that the loci are independent.

### 2.1. The variance of a single sample from a species

Consider the contribution of a single locus to the trait $X$ in species $i$. We proceed by first computing the variance of this measurement by conditioning on the random genealogy underlying that locus, and then average over all possible genealogies at the locus. In this context, the only thing that matters is the overall height of the genealogy. Given the genealogy at the locus, $G_l$, we have:

$$\mathrm{Var}(X_{i,l}|G_l) = \mu\sigma_M^2 T_{\mathrm{MRCA},l} \tag{S.12}$$

where $T_{\mathrm{MRCA},l}$ is the (random) *coalescence* time of the most recent common ancestor (MRCA) at this locus. Note that $T_{\mathrm{MRCA},l}$ occurs in the ancestral population of all species (see **Appendix 1—figure 1** for an example of the three species case).

(a)

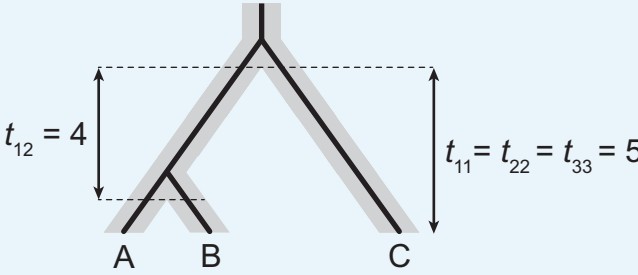

(b)

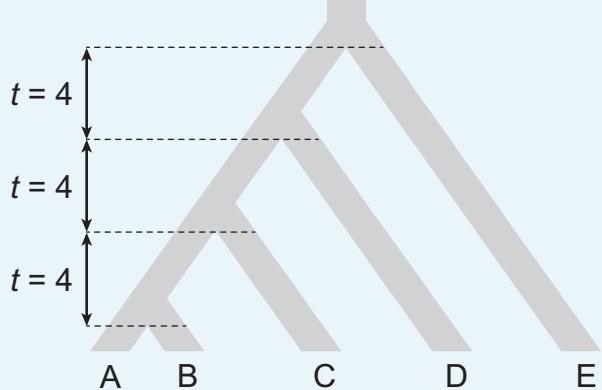

**Appendix 1—figure 1.** Divergence times, $t_{ij}$ ($t_{AB}$, $t_{(AB)C}$ and $t_{\mathrm{MRCA}}$ for the species tree depicted in gray), and coalescence times, $T_{ij,l}$ ($T_{AB,l}$, $T_{(AB)C,l}$ and $T_{\mathrm{MRCA},l}$ for genealogy $l$ depicted in black).
DOI: https://doi.org/10.7554/eLife.36482.013

Let $p_k$ be the probability that $k$ lineages enter the population of the MRCA of *all species*. (Note that for the three-species phylogeny, $p_2 = (1 - e^{-t/2N})$ and $p_3 = e^{-t/2N}$, which are defined in **Equations S.3 and S.6**).

Then, we use the law of total variance to get:

$$
\begin{aligned}
\mathrm{Var}(X_{i,l}) &= E(\mathrm{Var}(X_{i,l}|G_l)) + \mathrm{Var}(E(X_{i,l}|G_l)) \\
&= E(\mathrm{Var}(X_{i,l}|G_l)) \\
&= E(\mu\sigma_M^2 T_{\mathrm{MRCA},l}) \\
&= \mu\sigma_M^2 \left( t_{\mathrm{MRCA}} + 4N \sum_{k=2}^{n} \left(1 - \tfrac{1}{k}\right)p_k \right) \\
&= \mu\sigma_M^2 \left( t_{\mathrm{MRCA}} + 4N - 2N \sum_{k=2}^{n} \tfrac{2}{k} p_k \right)
\end{aligned} \tag{S.13}
$$

where $t_{\mathrm{MRCA}}$ is the time to the root of the species tree (**Appendix 1—figure 1**). The second line follows because we set the ancestral value to 0 and have no directionality to the mutational effects. In the fourth line, the total height of the genealogy, $T_{\mathrm{MRCA},l}$, is expressed as the time to the root of the species tree, $t_{\mathrm{MRCA}}$, plus the expected height of a genealogy whose $k$ lineages enter the MRCA of all species, $4N \sum_{k=2}^{n} \left(1 - \tfrac{1}{k}\right)p_k$. The second term of this sum follows because $4N(1 - 1/k)$ is the expected time to the MRCA for a sample of size $k$ from a

population of size $2N$ (and we sum over all possible probabilities $p_k$ that $k$ lineages enter the population of the MRCA of all species).

Computing $p_k$ can be done through a dynamic programming algorithm, and it depends only on the overall species tree, that is, it is not specific to any taxon.

Finally, we use this in the formula for the total variance of trait $X$ to get:

$$\mathrm{Var}(X_i) = L\mu\sigma_M^2\left(t_{\mathrm{MRCA}} + 4N - 2N\sum_{k=2}^{n}\frac{2}{k}p_k\right). \tag{S.14}$$

Given the same phylogeny, **Equations S.14 and S.3** evaluate to the same result.

## 2.2. The covariance between samples from two species

Now consider the covariance between a single sample from each of the two species. Again, we first condition on the genealogy underlying locus $l$, $G_l$, and then average over it to find the contribution to the variance. We have:

$$\mathrm{Cov}(X_{i,l}, X_{j,l}|G_l) = \mu\sigma_M^2\left(T_{\mathrm{MRCA},l} - T_{ij,l}\right), \tag{S.15}$$

where $T_{ij,l}$ is the *coalescence* time of the common ancestor of the lineages from species $i$ and the sample from species $j$ (e.g. $T_{AB,l}$ in **Appendix 1—figure 1**). We again denote by $p_k$ the probability that $k$ lineages enter the population of the MRCA of *all species*. We then use the law of total covariance to get:

$$\begin{aligned}
\mathrm{Cov}(X_{i,l}, X_{j,l}) &= E\left(\mathrm{Cov}(X_{i,l}, X_{j,l}|G_l)\right) + \mathrm{Cov}\left(E(X_{i,l}), E(X_{j,l})\right) \\
&= E\left(\mathrm{Cov}(X_{i,l}, X_{j,l}|G_l)\right) \\
&= \mu\sigma_M^2 E\left(T_{\mathrm{MRCA},l} - T_{ij,l}\right) \\
&= \mu\sigma_M^2\left[\left(t_{\mathrm{MRCA}} + \sum_{k=2}^{n}4N\left(1 - \frac{1}{k}\right)p_k\right) - (t_{ij} + 2N)\right] \\
&= \mu\sigma_M^2\left(t_{\mathrm{MRCA}} - t_{ij} + 2N - 2N\sum_{k=2}^{n}\frac{2}{k}p_k\right)
\end{aligned} \tag{S.16}$$

with $t_{ij}$ being the time of divergence between species $i$ and $j$ (e.g. $t_{AB}$ in **Appendix 1—figure 1**). The fourth line follows from replacing $T_{\mathrm{MRCA},l}$ and $T_{ij,l}$ with their expectations.

Once again, we can substitute the last line into the formula for the total covariance to get:

$$\mathrm{Cov}(X_i, X_j) = L\mu\sigma_M^2\left(t_{\mathrm{MRCA}} - t_{ij} + 2N - 2N\sum_{k=2}^{n}\frac{2}{k}p_k\right). \tag{S.17}$$

Note that in **Equation S.16 and (S.17)**, we compute the trait covariance between any pair of species (for any species tree) without enumerating the individual contributions and probabilities of each and all possible genealogies under the species tree. This is possible because all genealogies having coalescences before $t_{\mathrm{MRCA}}$ are jointly (and implicitly) dealt with by the recursive computation of $p_k$, which we do not lay out here for the sake of brevity.

Finally, given phylogeny ((A,B),C), and letting species $i = \mathrm{A}$, **Equation S.17** evaluates to the same results as **Equations S.6 and S.7** for $j = \mathrm{B}$ and $j = \mathrm{C}$, respectively.

## 2.3. A comparison of the covariance structure under the Brownian motion and coalescent models

Now, let us examine how the covariance structure derived above relates to the covariance that is usually assumed under a Brownian motion (BM) model of trait evolution. We first show that the variance-covariance matrix under the coalescent model can be represented in terms of the Brownian variance-covariance matrix. We then provide bounds that reveal important properties of the impact of incomplete lineage sorting (ILS). Finally, we explore the asymptotic

behavior of the variance-covariance matrix when one approaches no internal coalescences, that is, when all coalescences occur in the MRCA of all lineages.

Let $L\mu\sigma_M^2 = \sigma^2$ ($\sigma^2$ is the evolutionary rate in the BM model), to see that $L\mu\sigma_M^2 t_{\mathrm{MRCA}} = \sigma^2 t_{\mathrm{MRCA}}$ is the variance of a single species under BM with rate $\sigma^2$, and that $L\mu\sigma_M^2(t_{\mathrm{MRCA}} - t_{ij}) = \sigma^2(t_{\mathrm{MRCA}} - t_{ij})$ is the covariance between two species under BM with rate $\sigma^2$. This shows that there is a component of the covariance that is identical to a Brownian model, assuming rate $\sigma^2$. We can then combine our results from the previous two sections to see that the $ij$th entry of the variance-covariance matrix under the coalescent model is:

$$\Sigma_{ij} = \Sigma_{ij}^{(BM)} + \left(2N(1+\delta_{ij})L\mu\sigma_M^2 - 2NL\mu\sigma_M^2\sum_{k=2}^{n}\frac{2}{k}p_k\right), \tag{S.18}$$

where $\Sigma_{ij}^{(BM)}$ is the covariance under Brownian motion, and $\delta_{ij}$ is Kronecker's delta. Term $\left(2N(1+\delta_{ij})L\mu\sigma_M^2 - 2NL\mu\sigma_M^2\sum_k\frac{2}{k}p_k\right)$ is the contribution of ILS relative to $\Sigma_{ij}^{(BM)}$, which affects all entries of $\Sigma_{ij}$ equally. Interestingly, in this derivation no term indicating which trees contribute to each $\Sigma_{ij}$ entry (or how likely these trees are) is necessary.

## 2.3.1. The covariance structure of the BM and coalescent models in the limiting cases of no ILS vs. maximum ILS

What can be determined about the limiting cases of (i) all sister lineages sorting in their MRCA (no ILS), and (ii) all coalescent events occurring in the MRCA of all taxa (maximum ILS)?

First, note that $\sum_{k=2}^{n}\frac{2}{k}p_k = E\left(\frac{2}{K}\right)$, where $K$ is the random number of lineages that enter the MRCA population. Letting $E(K)$ be the expected number of lineages that enter the most recent common ancestor population, Jensen's inequality shows that:

$$E\left(\frac{2}{K}\right) \geq \frac{2}{E(K)}$$
$$\geq \frac{2}{n} \tag{S.19}$$

because $E(K) \leq n$. This is a tight lower bound on $\sum_{k=2}^{n}\frac{2}{k}p_k$ because when there are no coalescences until the MRCA population of all samples, $p_n = 1$ and $p_k = 0$ for $2 \leq k \leq n-1$, and in that case:

$$\sum_{k=2}^{n}\frac{2}{k}p_k = \frac{2}{n}. \tag{S.20}$$

On the other hand, observe that:

$$\sum_{k=2}^{n}\frac{2}{k}p_k \leq \sum_{k=2}^{n}\frac{2}{2}p_k$$
$$= \sum_{k=2}^{n}p_k \tag{S.21}$$
$$= 1.$$

This is a tight upper bound because when all coalescences happen during the internal branches of the species tree (i.e. no ILS), $p_2 = 1$ and $p_k = 0$ for $3 \leq k \leq n$, so:

$$\sum_{k=2}^{n}\frac{2}{k}p_k = 1. \tag{S.22}$$

Thus, we see that:

$$(\mathrm{Maximum\,ILS})\frac{2}{n} \leq \sum_{k=2}^{n}\frac{1}{k}p_k \leq 1(\mathrm{no\,ILS}).$$

Using these facts, we we can bound:

$$(\text{No ILS})\,\Sigma_{ij}^{(BM)} + 2N\delta_{ij}L\mu\sigma_M^2 \leq \Sigma_{ij} \leq \Sigma_{ij}^{(BM)} + 2NL\mu\sigma_M^2\left(1 - \frac{2}{n} + \delta_{ij}\right)(\text{Maximum ILS}).$$

Together, these results reveal several important aspects about the impact of ILS. First, it shows that when there is no ILS, $2N\left(1 + \delta_{ij}\right)L\mu\sigma_M^2$ is cancelled out by $2NL\mu\sigma_M^2\sum_k \frac{2}{k}p_k$, and $\Sigma_{ij}$ reduces to $\Sigma_{ij}^{(BM)}$ for $i \neq j$ (see main text for a simple worked example). The diagonal entries of $\Sigma_{ij}$ (i.e. the trait variances in different species), however, will be larger than the corresponding entries of $\Sigma_{ij}^{(BM)}$ *even in the absence of ILS*. Second, as long as there is any ILS, covariances will be increased relative to Brownian motion. And importantly, because the off-diagonal terms that are added to $\Sigma_{ij}^{(BM)}$ are independent of $i$ and $j$, we see that the impact of ancestral polymorphism cannot be modeled by simply changing the rate of Brownian motion.

### 2.3.2 Asymptotic behavior when internal coalescence is rare

When the frequency of internal coalescence approaches zero, the variance-covariance matrix converges to a matrix where all diagonal entries are identical and all off-diagonal entries are identical (convergence can be seen for up to 60% discordance in *Figure 5b*, main text). Here, we derive the form of the limiting variance-covariance matrix.

Consider that no internal coalescence results in $p_n = 1$ and $p_k = 0$ for $2 \leq k \leq n - 1$, so:

$$\sum_{k=2}^{n} \frac{2}{k}p_k = \frac{2}{n}. \tag{S.23}$$

So then,

$$\begin{aligned}
\Sigma_{ij} &= \Sigma_{ij}^{(BM)} + 2N\left(1 + \delta_{ij}\right)L\mu\sigma_M^2 - 2NL\mu\sigma_M^2\frac{2}{n} \\
&= \Sigma_{ij}^{(BM)} + 2NL\mu\sigma_M^2\left(1 - \frac{2}{n} + \delta_{ij}\right) \\
&\sim 2NL\mu\sigma_M^2\left(1 - \frac{2}{n} + \delta_{ij}\right),
\end{aligned} \tag{S.24}$$

where the asymptotics follow because in order for there to be very low amounts of internal coalescence, $N$ must be extremely large compared to any of the internal branch lengths. Thus, the Brownian component of the covariance is negligible.

This formula shows that even in the regime of maximal ILS, in which all coalescences happen in the MRCA of all species and there is no phylogenetic signal, the data are still correlated. This is because in this regime, every pair of lineages will be subtended by the same total (i.e. over all genealogies) expected branch length path, and as a result will share the same non-zero correlation. This pattern is not possible under Brownian motion, where lineages from a star phylogeny will be independent and identically distributed.

