## [Decision Letter]

Thank you for submitting your article "Evolutionary inferences about quantitative traits are affected by underlying genealogical discordance" for consideration by *eLife*. Your article has been reviewed by Patricia Wittkopp as the Senior Editor, a Reviewing Editor, and three reviewers. The following individuals involved in review of your submission have agreed to reveal their identity: Matthew Pennell (Reviewer #1).

Summary:

The paper by Mendes and colleagues develops a multispecies coalescent model for quantitative traits that takes into account genealogical discordance and how this affects trait evolution inferences. The study is an important conceptual contribution to the field of trait evolution. Typically, the field has taken an inferred phylogeny as a given without always being clear about whether the tree represents a species tree, gene tree, average tree, etc. Thinking about how gene tree variation should affect our understanding of trait evolution and the development of a model is an important step in this area.

Essential revisions:

1) The reviewers appreciate the novelty of a quantitative trait model based on the multi-species coalescent (MSC). As you describe throughout the paper, it has a number of properties that make it behave fundamentally differently than Brownian motion (BM). Such a model is definitely worth considering as an alternative to BM and furthermore allows for a more explicit linking between macro- and micro-evolution. This builds nicely on the recent work by Josh Schraiber (a co-author on this paper) and Michael Landis on how we might develop phylogenetic models for traits with different genetic architectures. We also really liked the partitioning of the un-modeled variance (captured by λ) into that caused by deep coalescence and that caused by ILS. This is very clever and well-thought out. However, we think it is somewhat misleading as it is currently framed. You use this the MSC trait model to support your claim that a) ILS can generate hemiplasy in quantitative traits (a cool finding, which we think you have proven) and b) this may - or in some parts of the paper you seem to suggest it is likely to - distort inferences from phylogenetic comparative methods (which we don't think you have). Our reasoning is that your comparison of BM with the MSC model rests on the assumption that BM is literally genetic drift (such that *σ^2^* is equal to G/Ne under the Lande model or, perhaps more plausibly, 2 times the mutational variance which you have). And if this is the case, then yes - failing to consider ILS - may indeed cause rates of trait evolution to appear inflated. However, there is abundant evidence (e.g., Lynch, 1991; Estes and Arnold, 2007; Harmon et al., 2010; Uyeda et al., 2011; etc.) which suggests macroevolutionary rates are much too slow to be compatible with drift in a strict sense. Even where traits are well described by a BM process, the parameter estimates make no sense when translated to a quantitative genetic interpretation of drift.

So why does this matter? Consider the case (as you do) of the phylogenetic ANOVA. The evolutionary scenario that is assumed by this statistical model is that some unobserved trait which has evolved along a phylogeny by some process (e.g., BM) and is influencing the distribution of both X and Y such that there is phylogenetic signal in the residual variation (see Pannell's recent paper with Josef Uyeda and Rosana Zenil-Ferguson on bioRxiv for more on this). Incorporating the phylogeny is not correcting the degrees of freedom due to the non-independence of data (as you write) but adjusting for non-independence of the residuals. If we were to take the BM == drift argument literally and apply a phylogenetic ANOVA this to a phylogeny spanning, say, tens or hundreds of millions of years, a back of the envelope calculation using plausible parameters for population sizes and genetic variances would suggest that this unobserved trait would likely vary many orders of magnitude across the tree. If we failed to consider or measure such a hypervariable (and presumably biologically important) trait that was directly influencing both X and Y in our ANOVA, we would have much bigger problems than ILS! As such, we think the comparisons that you use to draw your conclusions that hemiplasy affects comparative tests rests on some rather implausible scenarios. In order for us to be convinced that this indeed a problem, we would like to see a comparison of the expected variances and covariances generated under the MSC model using plausible sets of parameter values to variances and covariances that actually comparative datasets contain (i.e., a test of model adequacy).

But we also don't think the paper needs to rest on the premise that this misleads comparative tests. There is a lot of creative, interesting stuff here that substantially advance macroevolution and phylogenetics. These can stand alone as major contributions without the "this breaks things" angle, so we strongly suggest considering this alternative framing.

2) We recommend to accentuate co-estimating Pagel's λ (or any measure of phylogenetic signal correction) with a phylogenetic comparative model. The authors correctly point to this as a band-aid solution to discordance, but it is also a readily available and easily implemented correction while more sophisticated methods are developed. This practice is not as widely used as one would expect due to it's unclear meaning and this work would somewhat salvage its value.

3) Please provide a clearer explanation of how the coalescent variation and co-variation parameters map onto the phylogeny. It was not immediately clear what the difference between *t_e_*and *t* were. Furthermore, it would be helpful to have the terms within the equations broken down into conceptual chunks within the main text. For example, in equation 2 there are 3 main terms separated from one another by addition. What does (1- *e^-t/2N^) (t/2N* + 1) correspond to in a population? Although, it seems important that as the ratio of *t/2N* approaches 0 (large populations) and that this middle term drops out as 0; it is difficult to directly interpret what this means conceptually without background knowledge.

4) In the discussion of the most frequent gene tree (Introduction), the authors should point out that trait evolution studies usually involve much larger trees than those considered in this paper. Consequently, most or even all gene tree topologies are often distinct, and the concept of the most frequent gene tree is not very useful; so many trees are possible it might even be unlikely that a gene tree matching the species tree shows up at all, even if most gene trees have most of the clades in the species tree. We understand that for purposes of this paper, focusing on small trees is useful for gaining theoretical insight to what is going on. Unfortunately, some of the assumptions used also are less tenable for the large trees that are typically used in trait evolution studies, such as a molecular clock and constant population sizes, and the authors should make readers aware of these issues.

5) In the simulation study, are trait inferences made assuming the correct species tree? It might be pointed out that in actual trait evolution studies, using an only approximately estimated species tree which is incorrect is a further potential source of error – this is usually overlooked in the trait evolution literature.

6) Given all of the above, please consider revising your title.

---

## [Author Response]

Essential revisions:1) The reviewers appreciate the novelty of a quantitative trait model based on the multi-species coalescent (MSC). As you describe throughout the paper, it has a number of properties that make it behave fundamentally differently than Brownian motion (BM). Such a model is definitely worth considering as an alternative to BM and furthermore allows for a more explicit linking between macro- and micro-evolution. This builds nicely on the recent work by Josh Schraiber (a co-author on this paper) and Michael Landis on how we might develop phylogenetic models for traits with different genetic architectures. We also really liked the partitioning of the un-modeled variance (captured by λ) into that caused by deep coalescence and that caused by ILS. This is very clever and well-thought out. However, we think it is somewhat misleading as it is currently framed. You use this the MSC trait model to support your claim that a) ILS can generate hemiplasy in quantitative traits (a cool finding, which we think you have proven) and b) this may – or in some parts of the paper you seem to suggest it is likely to – distort inferences from phylogenetic comparative methods (which we don't think you have). Our reasoning is that your comparison of BM with the MSC model rests on the assumption that BM is literally genetic drift (such that σ^2^ is equal to G/Ne under the Lande model or, perhaps more plausibly, 2 times the mutational variance which you have). And if this is the case, then yes – failing to consider ILS – may indeed cause rates of trait evolution to appear inflated. However, there is abundant evidence (e.g., Lynch, 1991; Estes and Arnold; 2007, Harmon et al., 2010; Uyeda et al., 2011; etc.) which suggests macroevolutionary rates are much too slow to be compatible with drift in a strict sense. Even where traits are well described by a BM process, the parameter estimates make no sense when translated to a quantitative genetic interpretation of drift.So why this matters? Consider the case (as you do) of the phylogenetic ANOVA. The evolutionary scenario that is assumed by this statistical model is that some unobserved trait which has evolved along a phylogeny by some process (e.g., BM) and is influencing the distribution of both X and Y such that there is phylogenetic signal in the residual variation (see Pannell's recent paper with Josef Uyeda and Rosana Zenil-Ferguson on bioRxiv for more on this). Incorporating the phylogeny is not correcting the degrees of freedom due to the non-independence of data (as you write) but adjusting for non-independence of the residuals.

We have corrected our description of this method:

“Within a linear model framework, the phylogenetic ANOVA also corrects for the inflation of degrees of freedom caused by the non-independence of trait value errors around the regression line (which can be estimated by looking at their residuals) — which results from the hierarchical nature of the phylogenetic relationships among species (Felsenstein, 1985; Garland et al., 1993; Uyeda et al., 2018).” (Subsection “Hemiplasy can increase the false positive rate in phylogenetic hypothesis testing”).

If we were to take the BM == drift argument literally and apply a phylogenetic ANOVA this to a phylogeny spanning, say, tens or hundreds of millions of years, a back of the envelope calculation using plausible parameters for population sizes and genetic variances would suggest that this unobserved trait would likely vary many orders of magnitude across the tree. If we failed to consider or measure such a hypervariable (and presumably biologically important) trait that was directly influencing both X and Y in our ANOVA, we would have much bigger problems than ILS! As such, we think the comparisons that you use to draw your conclusions that hemiplasy affects comparative tests rests on some rather implausible scenarios. In order for us to be convinced that this indeed a problem, we would like to see a comparison of the expected variances and covariances generated under the MSC model using plausible sets of parameter values to variances and covariances that actually comparative datasets contain (i.e., a test of model adequacy).

We have significantly altered and expanded the paragraph that mentioned the equivalency of BM to an evolutionary model for a trait under drift (and included the references cited by the reviewer). We attempted to clarify that our results do not rest on the assumption of BM results needing to be interpreted in a “no selection” context – in fact, whether or not BM allows for selection is not particularly relevant for the point we try to make. In addition, we picked it because it’s the most studied, tractable model out there, and even models with usually better fit are based on it. We also recognize and now also mention the mismatch between the macroevolutionary scale and the scale of drift in larger trees.

“Although there are multiple possible interpretations of the underlying microevolutionary dynamics that lead to BM, we compare our model to it because it is tractable and popular, thus providing a clear baseline that is likely to be informative about the behavior of our model.” (Subsection “Characterizing trait distributions in the three-species case under the coalescent and Brownian motion models”).

“We have focused the comparisons of trait-value expectations from our model (which assumes no selection) to those under BM. The latter can be seen as equivalent to a quantitative genetics model in which many genes have small effects on a selectively unconstrained character (Lande, 1976; Harmon, 2017). […] Finally, when analyzing our simulated data here we employed the true, known species tree used in the simulations, but in typical phylogenetic analyses of real data an estimate of the phylogeny is used. These facts will certainly cause results from real-world studies to deviate from expectations.” (Discussion section).

We also point to how selection could impact our model results:

“Although many quantitative traits may be under selection, the selection coefficient on an individual allele is proportional its effect size (Keightley and Hill 1988). This implies that loci with small effects will experience very small selection coefficients, and that neutral expectations for genealogical quantities should still be justified.” (Subsection “Characterizing trait distributions in the three-species case under the coalescent and Brownian motion models”).

But we also don't think the paper needs to rest on the premise that this misleads comparative tests. There is a lot of creative, interesting stuff here that substantially advance macroevolution and phylogenetics. These can stand alone as major contributions without the "this breaks things" angle, so we strongly suggest considering this alternative framing.

We agree with the reviewers and have attempted to better convey the constructive aspects of our work. We modified the Abstract, which now reads:

“We present a multispecies coalescent model for quantitative traits that allows for evolutionary inferences at both micro- and macroevolutionary scales. A major advantage of this model is its ability to incorporate genealogical discordance underlying a quantitative trait. We show that discordance causes a decrease in the expected trait covariance between more closely related species relative to more distantly related species. If unaccounted for, this outcome can lead to an overestimation of a trait’s evolutionary rate, to a decrease in its phylogenetic signal, and to errors when examining shifts in mean trait values. The number of loci controlling a quantitative trait appears to be irrelevant to all trends reported, and discordance also affected discrete, threshold traits. Our model and analyses point to the conditions under which different methods should fare better or worse, in addition to indicating current and future approaches that can mitigate the effects of discordance.”

We have also edited passages from the Introduction, as well as added new ones, that hopefully minimize the destructive tone of our narrative, and also emphasize previous work linking micro and macroevolutionary-scale inferences:

“The realization that genealogical discordance is the rule rather than the exception (in species trees with short internal branches) has been followed closely by a growing awareness that failing to consider microevolutionary-scale processes in phylogenetic inferences can be problematic (Kubatko and Degnan, 2007; Edwards, 2009; Mendes and Hahn, 2016, 2018). In the context of species tree estimation, this has led to the development of popular methods that account for processes such as ILS and introgression (e.g., Liu, 2008; Than et al., 2008; Liu et al., 2009; Heled and Drummond, 2010; Larget et al., 2010; Bryant et al., 2012; Mirarab and Warnow, 2015; Claudia, Solís-Lemus and Ané, 2016). However, the development of models incorporating discordance in order to deal with trait evolution have lagged behind those estimating species trees.”

“In contrast, the manner in which genealogical discordance might affect studies of complex trait evolution is still not well understood. While past work has investigated how the genetic architecture of complex traits interacts with genetic drift to influence patterns of variation between populations and species (Lynch, 1988; Lynch, 1989; Lynch, 1994; Whitlock, 1999; Ovaskainen et al., 2011; Zhang et al., 2014), an interesting and still unanswered question is whether genealogical discordance can have an effect on these traits.”

“Here, we present a model of quantitative traits evolving under the multispecies coalescent that accounts for gene tree discordance, deriving the expected variances and covariances in quantitative traits under this model. We then apply traditional phylogenetic comparative methods to data simulated under the coalescent framework in order to better understand the impact of discordance on phylogenetic inference. Our framework makes it possible to vary levels of ILS and the number of loci controlling a quantitative trait (cf. Schraiber and Landis, 2015), and so we also address whether variation in genetic architecture has an effect on phylogenetic inference.”

As well as passages from the Discussion section:

“In the present study, we propose a multispecies coalescent model for quantitative trait evolution that incorporates the genealogical discordance that underlies such traits. We use this model to investigate whether discordance can affect phylogenetic inferences, mainly through the phenomenon of hemiplasy. We considered ILS to be the sole cause of gene tree discordance. By employing coalescent theory, we demonstrate that in the absence of ILS the coalescent and traditional BM model are equivalent with respect to the expected covariances between species trait values, but differ in terms of the species expected trait variances. In the presence of ILS, hemiplasy causes the expected covariance in trait values between pairs of more distantly related species to increase.”

2) We recommend to accentuate co-estimating Pagel's λ (or any measure of phylogenetic signal correction) with a phylogenetic comparative model. The authors correctly point to this as a band-aid solution to discordance, but it is also a readily available and easily implemented correction while more sophisticated methods are developed. This practice is not as widely used as one would expect due to it's unclear meaning and this work would somewhat salvage its value.

We moved the reference to using Pagel’s λ from the last paragraph of the paper to the paragraph where we discuss solutions to the problem, and also expanded its discussion:

“Another recommendation for avoiding the problems associated with discordance would be to use Pagel’s λ in analyses aimed at estimating evolutionary rates under the BM model. We have shown here that Pagel’s λ will “buffer” the effect of hemiplasy, minimizing the overall impact of genealogical discordance over the estimates.” (Discussion section).

3) Please provide a clearer explanation of how the coalescent variation and co-variation parameters map onto the phylogeny. It was not immediately clear what the difference between t_e_ and t were. Furthermore, it would be helpful to have the terms within the equations broken down into conceptual chunks within the main text. For example, in equation 2 there are 3 main terms separated from one another by addition. What does (1- e^-t/2N^) (t/2N + 1) correspond to in a population? Although, it seems important that as the ratio of t/2N approaches 0 (large populations) and that this middle term drops out as 0; it is difficult to directly interpret what this means conceptually without background knowledge.

We modified the definition of *t_e_* to read:

“*t_e_* is the length of terminal branch leading to species A and B (i.e., the length of the A and B leaves).” (Subsection “Brownian motion models”).

And added the whole paragraph below:

“The term within square brackets in equation 2 captures the total amount of time the trait has been evolving since the common ancestor of all lineages in the tree, which is equivalent to the total length of root-to-tip paths in all gene trees underlying it. The first term inside the brackets (*t_e_*) describes the length of the leaves, which are not affected by the coalescent process. The second term in the brackets computes the path length coming from gene trees in which the A and B lineages coalesce in their common ancestor (with probability 1 – *e^-t/2N^*), while the third term gives the path length coming from gene trees where A and B fail to coalesce in that ancestor (with probability *e^-t/2N^*). These two kinds of gene trees contribute path lengths proportional to (*t/2N* + 1) and (*t/2N* + 1 + 1/3), respectively. Note that the expected trait value variance within species C is given by a sum similar to equation 2, but that has the t/2N terms dropped from the path length component (and where *t_e_* corresponds to the terminal branch leading to species C; see section 1.1 in the Appendix).” (Subsection “Brownian motion models”).

4) In the discussion of the most frequent gene tree (Introduction), the authors should point out that trait evolution studies usually involve much larger trees than those considered in this paper. Consequently, most or even all gene tree topologies are often distinct, and the concept of the most frequent gene tree is not very useful; so many trees are possible it might even be unlikely that a gene tree matching the species tree shows up at all, even if most gene trees have most of the clades in the species tree. We understand that for purposes of this paper, focusing on small trees is useful for gaining theoretical insight to what is going on. Unfortunately, some of the assumptions used also are less tenable for the large trees that are typically used in trait evolution studies, such as a molecular clock and constant population sizes, and the authors should make readers aware of theseissues.

We now address the assumptions and limitations of our model more explicitly:

“Similarly, the assumptions used by our model about constant population sizes and mutation rates are only required to be true across phylogenetic knots, where the relevant coalescent events occur. This may be only a very small fraction of the total amount of time represented by large trees. We do not expect violations of these assumptions elsewhere in the tree — outside of knots — to have a qualitative effect any different than they would in non-coalescent models. Of course, the larger the periods of time covered by any such analysis, the harder it can be to infer parameters of interest, as in those circumstances even estimating phylogenies can be a daunting task. In such cases, ILS may be the least of a phylogeneticist’s concerns.” (Discussion section).

5) In the simulation study, are trait inferences made assuming the correct species tree? It might be pointed out that in actual trait evolution studies, using an only approximately estimated species tree which is incorrect is a further potential source of error – this is usually overlooked in the trait evolution literature.

We now point out that we used the correct species tree, and that in empirical studies, this is usually not possible:

“Finally, when analyzing our simulated data here we employed the true, known species tree used in the simulations, but in typical phylogenetic analyses of real data an estimate of the phylogeny is used. These facts will certainly cause results from real-world studies to deviate from expectations.” (Discussion section).

6) Given all of the above, please consider revising your title.

As requested, we have now modified our title to “A multispecies coalescent model for quantitative traits”.